# Assessment of Food Sources and the Intake of the Colourless Carotenoids Phytoene and Phytofluene in Spain

**DOI:** 10.3390/nu13124436

**Published:** 2021-12-11

**Authors:** Begoña Olmedilla-Alonso, Ana M. Benítez-González, Rocío Estévez-Santiago, Paula Mapelli-Brahm, Carla M. Stinco, Antonio J. Meléndez-Martínez

**Affiliations:** 1Department of Metabolism and Nutrition, Institute of Food Science, Technology and Nutrition (ICTAN-CSIC), C/José Antonio Novais, 10, 28040 Madrid, Spain; 2Food Colour and Quality Laboratory, Facultad de Farmacia, Universidad de Sevilla, 41012 Sevilla, Spain; abenitez@us.es (A.M.B.-G.); pmapelli@us.es (P.M.-B.); cstinco@us.es (C.M.S.); 3Faculty of Experimental Science, Universidad Francisco de Vitoria, 28223 Madrid, Spain; Rocio.estevez@ufv.es

**Keywords:** phytoene, phytofluene, food composition, carotenoid sources, carotenoid intake, Mediterranean diet

## Abstract

Phytoene (PT) and phytofluene (PTF), colorless carotenoids, have largely been ignored in food science studies, food technology, and nutrition. However, they are present in commonly consumed foods and may have health-promotion effects and possible uses as cosmetics. The goal of this study is to assess the most important food sources of PT and PTF and their dietary intakes in a representative sample of the adult Spanish population. A total of 62 food samples were analyzed (58 fruit and vegetables; seven items with different varieties/color) and carotenoid data of four foods (three fruits and one processed food) were compiled. PT concentration was higher than that of PTF in all the foods analyzed. The highest PT content was found in carrot, apricot, commercial tomato juice, and orange (7.3, 2.8, 2.0, and 1.1 mg/100 g, respectively). The highest PTF level was detected in carrots, commercial tomato sauce and canned tomato, apricot, and orange juice (1.7, 1.2, 1.0, 0.6, and 0.04 mg/100 g, respectively). The daily intakes of PT and PTF were 1.89 and 0.47 mg/person/day, respectively. The major contributors to the dietary intake of PT (98%) and PTF (73%) were: carrot, tomato, orange/orange juice, apricot, and watermelon. PT and PTF are mainly supplied by vegetables (81% and 69%, respectively). Considering the color of the edible part of the foods analyzed (fruit, vegetables, sauces, and beverages), the major contributor to the daily intake of PT and PTF (about 98%) were of red/orange color.

## 1. Introduction

Carotenoids are ancient and widespread isoprenoids biosynthesized by photosynthetic organisms (cyanobacteria, algae, plants) as well as some fungi, bacteria, and a few known invertebrate animals [1]. Research on these compounds in the agro-food and health fields continues to grow due to their versatility and usefulness in developing healthy foods and other related products including functional foods, nutraceuticals, supplements, and novel foods [2]. Beyond their role as pigments and precursors of vitamin A (some of them), there is ample evidence that carotenoids can spark health-promoting biological actions contributing to a decrease in the risk of cancer, cardiovascular disease, and bone, skin, or eye disorders [3,4]. There is also increasing evidence that they can have positive effects on mental and metabolic health, during pregnancy and early life [5,6]. There is also a renewed interest in the use of carotenoids for cosmetics [7,8].

Phytoene (PT) is the precursor of the vast majority of carotenoids. The formation of phytoene from geranylgeranyl pyrophosphate (GGPP) is catalyzed by phytoene synthase, whereas phytoene desaturase introduces new double bonds to form phytofluene (PTF) and then other more unsaturated compounds [1]. Very high quantities of PT can be obtained by using bleaching compounds (for instance, norflurazom) that block carotenoid biosynthesis at the level of the desaturation of carotenes early in the pathway [9]. Very high levels are also found in fruit color mutants, such as *Pinalate* or *Cara Cara* oranges, resulting from abnormal carotenoid biosynthesis [10,11]. Moreover, as it can be inferred from the foods analyzed for this study, important amounts of PT and PTF can be present in plant foods also accumulating other carotenoids as a result of “normal” carotenoid biosynthesis. However, the colorless carotenoids PT and PTF have been largely ignored in food science and technology, and nutrition studies. This has changed in recent years as studies and reviews conclude that they are among the major carotenoids in many foods and could be involved in health-promoting actions in humans [12,13,14,15,16,17]. Although more studies are now reporting on the level of colorless carotenoids in foods are this information can be found in the latest comprehensive databases on food carotenoid content [18,19,20], carotenoid intakes are mostly unknown as they have only been assessed in a Luxembourg study [15]. The goal of this study is to report on the level of colorless carotenoids in commonly consumed foods in Spain, assess their daily intakes, and identify the major food contributors.

## 2. Materials and Methods

### 2.1. Fruits, Vegetables and Processed Food Samples

Fruit and vegetables were acquired from supermarkets in Seville (Spain). A total of 62 samples were analyzed, 58 of which were fruits and vegetables, and four were ketchup, orange juice (from concentrate), tomato juice, and canned tomato. Different varieties/color of seven items were analyzed (guava, kiwi, lettuce, melon, peach, plum, pepper). Upon arrival to the laboratory, representative samples were chopped and freeze-dried. In the case of fruits, peel and the seeds were removed before processing.

### 2.2. Extraction and Rapid Resolution Liquid Chromatography (RRLC) Analysis of Carotenoids in Foods

50 mg samples (in triplicate) of homogenized freeze-dried powder were added 1 mL of extracting solvent (ethyl acetate/dichloromethane, 1:1 *v*/*v*) and then were vortexed. 700 µL of water was then added and centrifuged at 18.000× *g* for 5 min. After recovering the colored fraction, an additional 500 µL of dichloromethane was added and the mixture was vortexed and finally spun as described above. This operation was repeated until all color was gone. The pooled organic colored fractions were then evaporated to dryness in a vacuum concentrator (Eppendorf Concentrator Plus, Hamburg, Germany) and stored under N_2_ at −20 °C until analysis. A saponification step was added in the case of samples with high carotenoid ester content. All carotenoid extracts were saponified except for those from carrot, sweet potato, watermelon, tomato and green fruits, and vegetables. To that end, extracts were redissolved in 500 μL of dichloromethane and treated with 500 μL of methanolic KOH (30%, *w*/*v*) overnight in a nitrogen atmosphere, dim light, and at room temperature. Lastly, the organic phase was washed with NaCl 5% until rinse water pH was neutral and then concentrated to dryness. The dry residue was re-dissolved in 50–100 μL of acetonitrile before being injected into the RRLC system.

Extracts were analyzed by RRLC with UV/VIS diode array detector according to a routine method [21]. They were identified by comparing their chromatographic and UV/vis spectroscopic characteristics with those of standards. PT and PTF were isolated from appropriate sources in accordance with standard procedures [22]. External calibration was used for quantification. The concentration of each standard was measured spectrophotometrically according to the conditions described in Britton et al. [23]. Full standard curves were constructed with five different concentrations for each carotenoid in triplicate. Subsequently, they were injected into the RRLC in triplicate. Both curves showed good linearity (R^2^ = 0.999). The limit of detection ranged between 0.001 µg for PTF and 0.002 µg for PT, while the limit of quantification ranged from 0.002 µg to 0.007 µg for PTF and PT, respectively.

Extraction solvents (ethyl acetate and dichloromethane) were of analytical grade (VWR, Seattle, WA, USA). RRLC solvents, methanol (MeOH) and ethyl acetate, and acetonitrile were of RRLC grade and were acquired from Merck (Darmstadt, Germany). Water was purified in a NANOpure^®^ Diamond^TM^ system (Barnsted Inc., Dubuque, IO, USA).

### 2.3. Subjects

A representative sample of the adult Spanish population (*n* = 3000), aged 18–64 years, took part in the last Spanish National Dietary Survey (Spanish acronym ENIDE) conducted by the Spanish Agency for Food Safety and Nutrition (AESAN) in 2009 and 2010, to determine dietary intake patterns in the adult Spanish population. The ENIDE study considered demographic characteristics (gender, age, geographical zone, size of family, and level of education), lifestyle and diet, and seasonal consumption, analyzing the same number of surveys in each of the four seasons of the year. Food consumption data were obtained from a 24-h dietary recall completed by each participant with the aid of trained interviewers and a three-day diet diary which together constituted a record of over 12,000 days of dietary consumption [24]. The ENIDE survey used this information to show the daily consumption of foods (*n* > 400 items), classified into 12 groups according to intake. Each food (raw) was expressed in grams or mL (in the case of liquids) per person per day, considering the overall population (consumers and non-consumers). The ENIDE survey gathered information on the consumption of mixed meals/recipes, recording both the amounts consumed by the participants and the way in which the meals were prepared.

### 2.4. Dietary Carotenoid Intake Assessment

The mean of the food consumption data (g/person/day or ml/person/day) provided by the cross-section ENIDE survey [24] was used to assess the dietary intake of PT and PTF in the Spanish population. Data on the foods and amounts consumed were introduced into a specific software application for carotenoids [25] in which PT and PTF data content in 58 foods and 4 processed foods (orange juice from concentrate, ketchup, tomato: canned, juice) were analyzed in this study were included. PT and PTF concentrations in loquat, orange, pear, tomato sauce were obtained from the database of carotenoid content in Ibero-american foods [19]. PTF was not analyzed in loquat, orange, and pear, and PT was not analyzed in tomato sauce. The food groups included in the software are fruit, vegetables, oils and fats, snacks, nonalcoholic beverages, milk and dairy products, eggs and egg products, sauces, herbs, and spices. However, only data from foods belonging to the following groups were available: fruit, vegetables, sauces, non-alcoholic beverages. 

Foods considered in the calculations of the dietary intake of phytoene and phytofluene were (*n* = 66): (A) Fruits (*n* = 33, some of them of different varieties and colors): apple, apricot, avocado, banana, chestnut, flat peach, grapefruit, guava, guava (red), kaki, kiwi (green, yellow), lemon, loquat, mandarins, mango, melon (Cantaloupe), melon (white, yellow), nectarine, orange, orange juice (freshly squeeze), papaya, peach (gelo, igloo, red, yellow), pear, pineapple, plum (green, yellow), quince, watermelon. (B) Vegetables (*n* = 28, some of different varieties and colors): Artichoke, asparagus (green), beans (green), broccoli, cabbage, carrot, cauliflower, chard, corn, cucumber, eggplant, garlic (white), lamb’s lettuce, lettuce (“heart”, iceberg, romana), mushroom, peas, pepper (green, orange, red, yellow), potato, pumpkin, spinach, sweet potato, tomato, zucchini. (C) Juices and processed foods (*n* = 5): Ketchup, orange juice (from concentrate), tomato (canned, juice, sauce).

The PT and PTF content in the foods consumed was multiplied by the amount of edible portion ingested (g/person/day) and this provided an estimate of the contribution of the food to the intake of these carotenoids. Intakes from individual foods were added up to yield individual and total carotenoid intake, as well as their relative contribution. A series of assumptions were made regarding the intake of the following foods and subsequent calculation of the carotenoid: artichoke (raw plus canned), lamb’s lettuce plus watercress/2, asparagus (green and white)/2, kiwi (4/5 green and 1/5 yellow), lettuces (heart+ iceberg + romana/3), peach (yellow + gelo + iglo + red/4), melon (white + yellow + galo/3), orange juice (natural + concentrated), peppers (yellow + orange + red + green/4), grapefruit (pink + yellow/2), carrot (raw + boiled + canned), prepared sauces (for example, tomato sauce, Bolognese sauce) were quantified considering the mean value of the concentrations of tomato paste and tomato puree reported in the food composition table [16].

The major food sources of PT and PTF in the diet of the Spanish population were thus determined. The contribution made by the group comprised of fruit and vegetables to the intake of these carotenoids is presented both for individual items and items grouped by color (white/yellow, green, red/orange).

## 3. Results

### 3.1. Phytoene and Phytofluene Concentration in Spanish Foods

Table 1 shows the concentration of PT and PTF in foods consumed by the Spanish population. Colorless carotenoids were detected in 21 of the 66 foods used in the dietary PT and PTF dietary assessment. Colorless carotenoids were found in traditional Mediterranean diet foods (such as pear, peaches, grapefruit, mandarin, nectarine, quince, watermelon, and tomato derivatives) and others that could be considered non-traditional or exotic (guavas, loquat, papaya) (Figure 1). The concentration of PT was higher than that of PTF in all the foods assessed except for tomato sauce in which no PT was detected. The highest PT contents were found in carrot, apricot, commercial tomato juice, and orange (7.3, 2.8, 2.0, and 1.1 mg/100 g, respectively). PT concentration in carrots was four times higher than that of PTF. The highest PTF levels were detected in carrot, commercial tomato sauce, and canned tomato, apricot, and orange juice (1.7, 1.2, 1.0, 0.6, and 0.04 mg/100 g, respectively).

### 3.2. Daily Intakes of Phytoene and Phytofluene in Spain

The contribution of each fruit and vegetable to the dietary intake of PT and PTF in the Spanish diet is shown in Table 1. The daily total intake of PT and PTF is 1889.2 μg/p/day and 470.4 μg/day respectively. However, intake may be slightly higher as PT and PTF data from four foods (tomato sauce, orange, pear, loquat) were taken from the Ibero-american carotenoid database [19], and these foods were not analyzed for PTF or PT. Five foods are major contributors to the dietary intake of PT and PTF: carrot, tomato, orange/orange juice, apricot, and watermelon, which account for 98% and 73% of the dietary intake of PT and PTF, respectively. Figure 2A,B shows the major food contributors to the PT and PTF dietary intake.

Table 2 shows the contribution of the different food groups to their daily intake. The main source of these colorless carotenoids is vegetables accounting for 81% of PT and 69% of PTF intake. Fruit and non-alcoholic beverages (orange juice and tomato juice) supply 352.8 and 27 µg/p/d of PT and PFT, respectively, accounting for 18.7% and 5.8% of the daily intake of PT and PTF, respectively.

Table 3 shows the contribution of these foods (fruit, vegetables, sauces, and beverages) to the daily intake of PT and PTF grouped according to the color of their edible part. Red/orange colored foods are the major contributor to the daily intake of PT and PTF (about 98%), while white-yellowish foods account for 2% of the intake of the two carotenoids. Green foods do not supply these carotenoids.

## 4. Discussion

### 4.1. Phytoene and Phytofluene Concentrations in Spanish Foods

PT and PTF were found in only about one-third of the foods (fruit, vegetables, sausages, and beverages) from those habitually consumed in the adult Spanish population [24]. The highest PT concentrations (>1 mg/100 g) were found in carrot, apricot, tomato (fresh, canned, juice), and orange (fresh) (7.3, 2.8, 2.0, and 1.1 mg/100 g, respectively). Carrot, tomato (canned, juice), and apricot are also the foods with the highest PTF concentrations (1.7, 1.0, 0.9, 0.6 mg/100 g, respectively). PT concentration was higher than that of PTF in all the foods analyzed, which is consistent with other studies [26].

PT and PTF concentrations in carrots are quite higher than those reported in other studies (1.3–1.8 and 0.6 mg/100 g for PT and PTF, respectively) [19,25]. In contrast, those found in apricot coincided with the values reported elsewhere [19,25]. Among Mediterranean fresh fruits, apricot appears to be the richest in colorless carotenoids, the main carotenoids found in commercially available varieties of this fruit [18,26]. Concentration in oranges is higher for PT in fresh orange and in the range of PT and PTF concentrations published in other studies (0.05–0.56 mg PT/100 g and 0.04–0.18 mg PTF/100 g) as tabulated in a review [26]. PT concentration in orange juice was nearly nine times lower than that found in the fresh fruit. This could be due to the fact that a large part of the pulp (where the carotenoids are found) is removed from the juice and the carotenoids break down during juicing as this process damages cell structures which could contribute to a carotenoid loss [26]. No PT or PTF was detected in orange juice made from concentrate. In this case, the thermal treatments this product is subjected to could contribute to carotenoid degradation [27].

The fact that highly variable concentrations of colorless carotenoids have been found in the literature for the same food is not surprising as the secondary metabolites content of fruits and vegetables is dependent on many factors, including the genotype, climate, agronomic practices as well as technological and/or culinary practices, among others [7]. As an example, tomatoes of the same variety grown under the same conditions have been known to exhibit differences in PT and PTF content of four-fold and six-fold, respectively [28]. As for differences across tomato cultivars, a recent study showed that PT levels ranged from 0.3 (‘Cherry yellow’) to 252.6 mg/100 g dry weight (‘Orange’) (nearly 840-fold difference). PTF fluctuated between non-detectable levels and 12.3 mg/100 g dry weight (‘Orange’) [29]. There are also citrus mutants with surprisingly high colorless carotenoids levels, such as the color mutants Pinalate (yellow-fleshed) or Cara Cara (red-fleshed), whose PT and PTF levels reported in a previous study [30] could be categorized as very high and high, respectively. Moreover, agronomic factors such as reduction in nitrogen fertilization at different stages have been shown to have different effects on the biosynthesis of tomato carotenoids, including colorless carotenoids [31]. Regulated deficit irrigation can also significantly affect general carotenoid levels and PT and PTF in particular, with important genotype differences. Other factors such as cluster position or season also have an impact [32,33]. Industrial processes such as pasteurization (30 s at 90 °C) of Pinalate orange juice was likewise reported to reduce colorless carotenoid levels by approximately 68%, although in another study pasteurization (92 °C/30 s or 85 °C/15 s) did not have a significant effect on the total carotenoid concentration of ‘Lane late’ orange juice, whereas high-pressure homogenization (HPH, 150 MPa, 68 °C/15 s) decreased PT levels by 25% but PTF levels apparently increased by 10% [34].

Lastly, variations resulting from analytical procedures (sample treatment, carotenoid extraction, saponification, identification, quantitation) could also be expected. For instance, although PT and PTF are carotenes and are not esterified to fatty acids, saponification of the carotenoid extracts containing them can lead to an underestimation of their contents. As an example, recoveries of only 82% of β-carotene were reported for saponified extracts [21]. Moreover, the presence of some (typically 3–5) geometrical isomers of PTF normally present in foods [12], could complicate the quantification of this carotenoid and partially explain the great variability found.

The criteria proposed by Britton and Khachik [35] was applied to try to categorize the foods analyzed by PT and PTF levels. Food sources containing carotenoids are classified according to their content as low (0–0.1 mg/100 g), moderate (0.1–0.5 mg/100 g), high (0.5–2 mg/100 g), or very high (>2 mg/100 g). According to this criterion, carrots, apricots, and tomato juice have very high PT content and carrots, tomato sauce, and canned tomato have high PTF levels. Aside from carrot, tomato, tomato-based products, and apricot, orange was the only other food analyzed that had high PT and/or PTF levels with 1.1 mg of PT/100 g FW. However, PTF could not be detected in this fruit. In general, tomato and derivatives (ketchup, canned, paste, juice, soup, sauce, and puree) are well-known sources of colorless carotenoids with concentrations typically ranging from moderate to very high [26].

### 4.2. Phytoene and Phytofluene Dietary Intakes in Spanish Adult Population

The daily intake of PT and PTF was 1.9 and 0.5 mg/p/day, respectively. However, this intake could be slightly higher due to the lack of PT and/or PTF data on four of the foods that were not analyzed for these carotenoids and which data were taken from the Ibero-American carotenoid database [19]. This higher intake of PT (four times that of PTF) is also reflected in the concentrations found in the main contributors to dietary intake. Only five red/orange color (carrot, tomato, orange/orange juice, apricot, and watermelon) exhibited higher PT levels compared to PTF (three to five times higher). The PT and PTF dietary intake among the Spanish population is comparable to that reported for adults residing in Luxembourg recruited for an epidemiological cardio-vascular risk-factor study (2.0 mg/day for PT and 0.7 mg/day for PTF) [18]. In the case of Luxembourg, the ratio between PT and PTF was 3-fold [18], thus, lower than that found in this study. Both studies were conducted on a representative sample of the population (n = 3000 in Spain and n = 1432 in Luxembourg) of similar age ranges (18–64 y in Spain and 18–69 y in Luxembourg), during similar time periods (2009-10 in Spain and 2007-09 in Luxembourg) and similar dietary assessment methods (self-administered food frequency questionnaire) but using different food composition table of carotenoid. 

A comparison of the dietary intakes of PT and PTF with the other six main carotenoids we previously assessed in the Spanish population, with the same food consumption data and calculations for the carotenoid dietary intake (Figure 3) provided interesting information.: PT intake (1890 μg/p/d) is lower than that of lycopene (3056 μg/p/d), but higher than β-carotene (1459 μg/p/day), lutein plus zeaxanthin (1235 μg/p/day), β-cryptoxanthin (322 μg/p/day) and α-carotene (269 μg/p/d) [36,37]. The PTF intake (470 μg/p/d) is lower than that of non-provitamin A carotenoids (lycopene, lutein plus zeaxanthin) and of β-carotene, but higher than β-cryptoxanthin and α-carotene. In the Luxembourg study, PT intake (2000 μg/p/d) was higher than that of lycopene (1800 μg/day), lutein (1500 μg/day) β-cryptoxanthin (1400 μg/day), and zeaxanthin (300 μg/day) but lower than the intake of α-carotene plus β-carotene (7600 μg/p/d) [18].

Vegetables are the number one contributor to dietary intake of PT (81%) and PTF (69%) among the food group assessed (fruit, vegetables, sauces, non-alcoholic beverages). They are also major contributors to the dietary intake of non-provitamin A carotenoids (lycopene, lutein, and zeaxanthin) [37] and provitamin A carotenoids as a whole (α- and β-carotene, β-cryptoxanthin), but not of β-cryptoxanthin whose main source is fruit [36].

In terms of the color of the edible part (Table 3), red/orange foods are the major contributors to the dietary intake of PT and PTF (98%) and are also major contributors to the dietary intake of the provitamin A carotenoids [36]. More specifically, carrots and tomatoes are the major food contributors to PT, PTF, and provitamin A carotenoids intake. Red/orange foods contribute to the intake of lycopene (100%), α-carotene (95%), and β-cryptoxanthin (98.7%). However, both red/orange (68.8%) and green (24.4%) foods contribute to the β-carotene intake [36]. However, green foods do not contribute to PT or PTF intake.

Dietary intake of PT accounts for 21.6% of total carotenoid intake (PT, PTF, lutein, zeaxanthin, lycopene, α-carotene, β-carotene, β-cryptoxanthin) among the Spanish population: 1890 and 470 μg/p/d of PT and PTF, respectively and, 6397.2 μg/p/d of the other six carotenoids [37] resulting in a total intake of 8757.2 μg carotenoids/p/d. In the Luxembourg study, the total intake of carotenoids (including neoxanthin and violaxanthin intakes) was 14,300 μg/p/d, and the intake of PT accounted for 12% of that total, only slightly more than half of the ratio estimated for the Spanish population.

The presence of PT and PTF in human plasma, milk, and in several tissues at levels comparable to the other major carotenoids previously mention is already known [7,38,39]. Recently, it has been reported that they account for approximately 25% of total carotenoids in adipose tissue [14] and that they are readily detected in the feces of humans [40,41]. Concerning possible health-promoting biological actions of the colorless carotenoids, there is evidence of different nature indicating that they may be involved in antioxidant and anti-inflammatory, anticarcinogenic actions [42,43,44,45,46,47,48]. Thus, these colorless carotenoids could also help promote health as well as the colored carotenoids typically studied in this regard (lutein, zeaxanthin, β-cryptoxanthin, α-carotene, β-carotene, lycopene). One key difference is that PT and PTF absorb maximally in the UV region of the spectra, which makes them especially interesting in the protection against this radiation by dietary means. This can be harnessed for the biotechnological production of colorless-carotenoid rich products, for instance, a microalgae, considering that the biosynthesis of colored carotenoids can be blocked by chemical compounds [9].

Being the precursors of virtually all colored carotenoids, it might be hypothesized that PT and PTF levels and potential intakes could have been considerably higher before the advent of agriculture and then the domestication of crops based on favorable attributes including their color.

## 5. Conclusions

Colorless carotenoids are present in one-third of the plant-derived foods (fruits, vegetables, and derivatives) shown in this study from those habitually consumed in Spain, especially in traditional foods comprising the Mediterranean diet. PT intake is approximately four times that of PTF, consistent with the difference in their contents in the foods analyzed. The main sources of PT and PTF are carrot, apricot, tomato and tomato products, and orange. PT and PTF intake (1.89 and 0.47 mg/person/d) is similar to that reported among the Luxembourg population in a comparable study in terms of participant characteristics and methodology. Five foods are the major contributors to the dietary intake of PT and PTF: carrot, tomato, orange/orange juice, apricot, and watermelon, which account for 98% and 73%, respectively. Vegetables are the main source of colorless carotenoids and, considering the color of the edible part, red/orange foods account for approximately 98% of dietary intake while green colored foods make no contribution. Dietary intake of PT accounts for 21.6% of total carotenoid intake (lutein, zeaxanthin, lycopene, phytoene, phytofluene, α-carotene, β-carotene, β-cryptoxanthin). The results of this study again highlight the need to consider the possible role played by colorless carotenoids in promoting health.

## Figures and Tables

**Figure 1 nutrients-13-04436-f001:**
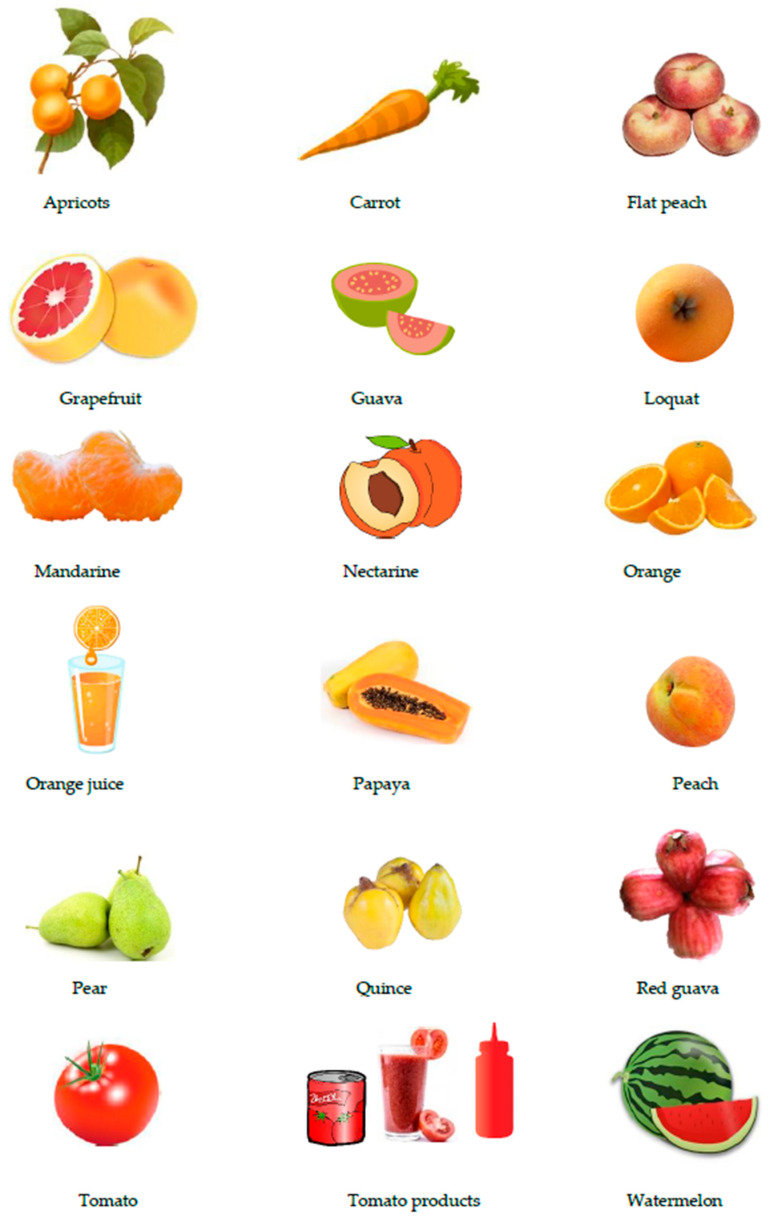
Some sources of colorless carotenoids in Spanish dietary intake.

**Figure 2 nutrients-13-04436-f002:**
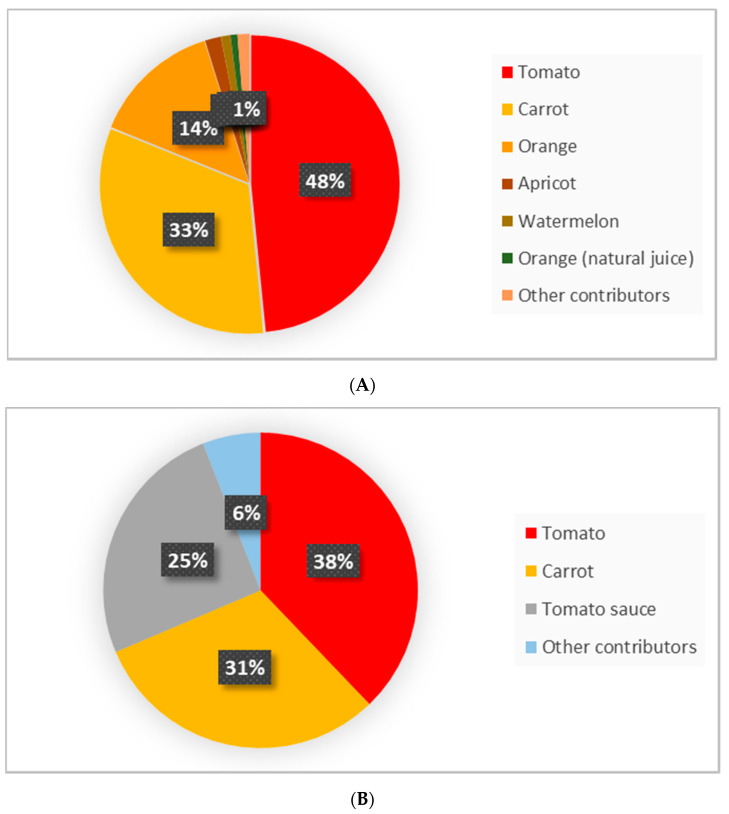
Major contributors to the dietary intake of phytoene (**A**) and phytofluene (**B**).

**Figure 3 nutrients-13-04436-f003:**
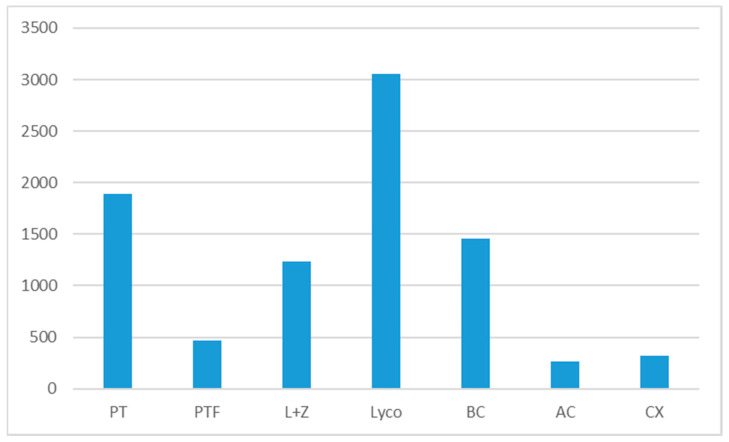
Carotenoids dietary intake in Spanish population (µg/p/day). Lutein plus zeaxanthin (L + Z) and lycopene (Lyco) from [37]. β-carotene (BC), α-carotene (AC) and β-cryptoxanthin (CX) from [36].

**Table 1 nutrients-13-04436-t001:** Phytoene and phytofluene content (µg/100 g fresh weight) (a) of fruits, vegetables and processed foods and daily intake (µg/person) and the consumption of fruits, vegetables, sauces and orange and tomato juices among the adult Spanish population.

Name	Spanish Name	Scientific Name	Edible Portion	Content (µg)/100 g Fresh Weight ^(a)^		Food Intake (g/p/day)	Dietary Intake (µg/day)	
				Phytoene	Phytofluene		Phytoene	Phytofluene
Fruits								
Apple	Manzana	*Malus domestica*	80			41.4	0.0	0.0
Apricot	Albaricoque	*Prunus armeniaca*	93	2818	616	1.3	33.0	7.2
Avocado	Aguacate	*Persea americana*	72			1	0.0	0.0
Banana	Plátano	*Musa × paradisiaca*	60			24.3	0.0	0.0
Chesnut	Castaña	*Castanea sativa*	82			0.2	0.0	0.0
Flat peach	Paraguaya	*Prunus persica var. platycarpa*	88	81	12	0.6	0.4	0.1
Grapefruit	Pomelo	*Citrus × paradisi*	68	151	6	0.2	0.2	0.0
Guava	Guayaba	*Psidium guajava*	89	454	83	0.005	0.0	0.0
Guava (red)	Guayaba (roja)	*Psidium guajava*	89	187	32	0.005	0.0	0.0
Kaki	Kaki	*Diospyros kaki*	87			0.6	0.0	0.0
Kiwi (green)	Kiwi (verde)	*Actinidia deliciosa*	66			5.7	0.0	0.0
Kiwi (yellow)	Kiwi (amarillo)	*Actinidia deliciosa*	66			1.4	0.0	0.0
Lemon	Limón	*Citrus × limon*	60			0.9	0.0	0.0
Loquat ^(b)^	Níspero	*Eriobotrya japonica*	65	26		0.1	0.0	
Mandarine	Mandarina	*Citrus reticulata*	73	60	51	9.8	4.3	3.7
Mango	Mango	*Mangifera indica*	68			0.4	0.0	0.0
Melon (cantaloupe)	Melón galo	*Cucumis melo var. reticulatus*	55			4.8	0.0	0.0
Melon (white)	Melón piel de sapo	*Cucumis melo ‘Santa Claus’*	62			4.8	0.0	0.0
Melon (yellow)	Melón (amarillo)	*Cucumis melo L.*	60			4.8	0.0	0.0
Nectarine	Nectarina	*Prunus persica var. nucipersica*	89	29	6	0.04	0.0	0.0
Orange ^(b)^	Naranja	*Citrus × sinensis*	72	1065		34.6	265.6	
Orange juice (freshly squeeze)	Naranja (zumo natural)	*Citrus × sinensis*	100	122	40	11.9	14.5	4.8
Papaya	Papaya	*Carica papaya*	75	12	10	0.9	0.1	0.1
Peach gelo	Melocotón gelo	*Prunus persica*	88			3.8	0.0	0.0
Peach (igloo)	Melocotón iglú	*Prunus persica*	88			3.8	0.0	0.0
Peach (red)	Melocotón (rojo)	*Prunus persica*	69	99	14	3.8	2.6	0.4
Peach (yellow)	Melocotón (amarillo)	*Prunus persica*	69	26	2	3.8	0.7	0.1
Pear ^(b)^	Pera	*Pyrus communis*	80	28.5		18.3	4.2	
Pineapple	Piña	*Ananas comosus*	57			6.7	0.0	0.0
Plum (green)	Ciruela (verde)	*Prunus domestica subsp. domestica*	85			1.0	0.0	0.0
Plum (yellow)	Ciruela (amarilla)	*Prunus domestica subsp. domestica*	92			1.0	0.0	0.0
Quince	Membrillo	*Cydonia oblonga*	61	116	44	0.4	0.2	0.1
Watermelon	Sandía	*Citrullus lanatus*	78	144	55	17.0	19.0	7.3
Subtotal						209.4	344.8	23.8
Vegetables								
Artichoke	Alcachofa	*Cynara scolymus*	47			2.6	0.0	0.0
Asparagus (green)	Espárrago (verde)	*Asparagus officinalis*	50			1.8	0.0	0.0
Beans (green)	Judías (verdes)	*Phaseolus vulgaris var. vulgaris*	93			8.3	0.0	0.0
Broccoli	Brécol	*Brassica oleracea var. italica*	97			0.1	0.0	0.0
Cabbage	Col	*Brassica oleracea*				2.3	0.0	0.0
Carrot	Zanahoria	*Daucus carota*	85	7264	1701	10.0	618.1	144.7
Cauliflower	Coliflor	*Brassica oleracea var. botrytis*	84			2.9	0.0	0.0
Chard	Acelga	*Beta vulgaris var. Cicla*	88			3.0	0.0	0.0
Corn	Maíz	*Zea mays*	100			1.8	0.0	0.0
Cucumber	Pepino	*Cucumis sativus*	70			4.6	0.0	0.0
Eggplant	Berenjena	*Solanum melongena*	85			3.4	0.0	0.0
Garlic (white)	Ajo (blanco)	*Allium sativum*	100			2.6	0.0	0.0
Lamb’s lettuce	Canónigo	*Valerianella locusta*	100			0.3	0.0	0.0
Lettuce (heart)	Lechuga (cogollo)	*Lactiva longifolia*	100			6.9	0.0	0.0
Lettuce (iceberg)	Lechuga (iceberg)	*Lactuca sativa var. Capitata*	88			6.9	0.0	0.0
Lettuce (romana)	Lechuga (romana)	*Lactuca sativa*	50			6.9	0.0	0.0
Mushroom	Champiñón	*Agaricus bisporus*	80			5.6	0.0	0.0
Peas	Guisante	*Pisum sativum*	100			3.5	0.0	0.0
Pepper (green)	Pimiento (verde)	*Capsicum annuum Group*	95			3.4	0.0	0.0
Pepper (orange)	Pimiento (naranja)	*Capsicum annuum Group*	87			3.4	0.0	0.0
Pepper (red)	Pimiento (rojo)	*Capsicum annuum Group*	85			3.4	0.0	0.0
Pepper (yellow)	Pimiento (amarillo)	*Capsicum annuum Group*	81			3.4	0.0	0.0
Potato	Patata	*Solanum tuberosum*	74			68.4	0.0	0.0
Pumpkin	Calabaza	*Cucurbita maxima*	67			2.2	0.0	0.0
Spinach	Espinaca	*Spinacia oleracea*	76			4.4	0.0	0.0
Sweet potato	Batata	*Ipomoea batatas*	100			0.1	0.0	0.0
Tomato	Tomate	*Solanum lycopersicum*	97	1697	330	55.6	915.2	178.0
Zucchini	Calabacín	*Cucurbita pepo*	79			6.3	0.0	0.0
Subtotal						224.1	1533.3	322.7
Processed Foods								
Orange juice (from concentrate)	Naranja (zumo a partir de concentrado)		100	0	0	11.9	0.0	0.0
Tomato canned	Tomate enlatado		100	1878	999	0	0.0	0.0
Tomato juice	Tomate en zumo		100	1994	884	0.4	7.8	3.4
Tomato sauce ^(b)^	Tomate en salsa		100		1165	10.3		119.6
Ketchup	Kétchup		100	391	99	1.0	4.0	1.0
Subtotal						23.6	11.8	124.0
TOTAL						457.1	1889.9	470.5

^(a)^ phytoene and phytofluene: empty boxes correspond to concentrations under detection limit. ^(b)^ data compiled from reference [19] and in these foods PT or PTF content were not analyzed.

**Table 2 nutrients-13-04436-t002:** Daily PT and PTF intake (μg/person/d) from the different food groups in the Spanish adult population.

Food Group	Phytoene	Phytofluene
Fruit	330.5	18.8
Vegetables	1533.3	322.7
Sauces	4.0	120.7
Non alcoholic beverages (orange juice, tomato juice)	22.3	8.2
Total	1890	470

**Table 3 nutrients-13-04436-t003:** Contribution of foods (fruit, vegetables, sauces, and beverages) grouped according to the color of their edible part to the daily intake (μg/person/d).

Food Color	Phytoene	Phytofluene
Red/orange	1852.0	463.0
Green	0.0	0.0
White-yellowish	37.9	7.4

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
