# Peer review of "Assessment of Food Sources and the Intake of the Colourless Carotenoids Phytoene and Phytofluene in Spain"

_nutrients, 2021, doi:10.3390/nu13124436_

Round 1
Reviewer 1 Report
The manuscript entitled “Assessment of food sources and the intake of the colourless carotenoids phytoene and phytofluene in Spain” from Olmedilla-Alonso et al. analyzed the sources of phytoene (PT) and phytofluene (PTF) in fruits, vegetables and processed food, mainly colored ones, as well as the daily the PT and PTF intake in the Spanish population.
The MS is well in the scope of Nutrients but needs some improvements:
- Material and methods: the title of the 2.1 indicated only fruits and vegetables, but the authors also analyzed processed foods such as tomato juice or ketchup. Please modify the title of the §2.1 to adequate to the content. In the same way, the Table 1 title could be also modified, as well as the organization of the topics: in my opinion it cannot be considered that tomato sauce or ketchup are vegetables. Please do three parts in the table: fruits, vegetables and processed foods.
- In the Introduction and other places in the text, the authors indicated the importance of the study regarding the scarcity of analyses related to colorless carotenoids, but at the same time, they focused on the analysis of color fruits, vegetables and processed foods. Why? Please explain this point in the material and methods or in the introduction. Please also discussed the implication on the carotenoid biosynthetic pathways. Does this mean that only when the whole biosynthetic pathway is activated, the PT and PTF could be high? Some color indicator (°Hue or chlorophyll/carotenoid index-CCI) could be added to the Table 1 to make a comparison/correlation with the PT and PTF content (in a more detailed way than the Table 3).
- Figure 3: Did the references 33 and 32 (used to make the figure) use the same analysis and quantification parameters? We believe that it is risky and inadequate to compare data from different works in a unique figure. See also comment 3 above about references to check the possibility to do this comparison and corresponding figure.
- The discussion could be improved to be more assertive and interesting for the reader (not only a comparison with previously local works like the one from Luxembourg). What is the biological importance of the study regarding the carotenoid biosynthetic pathway? Regarding the nutritional behavior of the Spanish population? Regarding nutritional and biotechnological aspects related to colorless vs colored carotenoids?
Author Response
The MS is well in the scope of Nutrients but needs some improvements:
- Material and methods: the title of the 2.1 indicated only fruits and vegetables, but the authors also analyzed processed foods such as tomato juice or ketchup. Please modify the title of the §2.1 to adequate to the content.
Reply: The title of the 2.1. now reads: Fruits, vegetables and processed food samples.
In the same way, the Table 1 title could be also modified, as well as the organization of the topics: in my opinion it cannot be considered that tomato sauce or ketchup are vegetables. Please do three parts in the table: fruits, vegetables and processed foods.
Reply: Table 1 has been modified and data are shown according three groups: fruits, vegetables and processed foods. Title of the table 1 now reads: “Phytoene and phytofluene content (µg/100g fresh weight)(a) of fruits, vegetables and processed foods ….”
- In the Introduction and other places in the text, the authors indicated the importance of the study regarding the scarcity of analyses related to colorless carotenoids, but at the same time, they focused on the analysis of color fruits, vegetables and processed foods. Why? Please explain this point in the material and methods or in the introduction. Please also discussed the implication on the carotenoid biosynthetic pathways. Does this mean that only when the whole biosynthetic pathway is activated, the PT and PTF could be high?
Reply: Phytoene and phytofluene are precursors of the vast majority of carotenoids and thus, can also be present in plant foods accumulating other carotenoids as a result of a normal carotenoid biosynthesis. The following sentences has been included in the introduction (lines 43-53): “Phytoene (PT) is the precursor of the vast majority of carotenoids. The formation of phytoene from geranylgeranyl pyrophosphate (GGPP) is catalyzed by phytoene synthase, whereas phytoene desaturase introduces new double bonds to form phytofluene (PTF) and then other more unsaturated compounds [1]. Very high quantities of phytoene can be obtained by using bleaching compounds (for instance norflurazom) that block carotenoid biosynthesis at the level of the desaturation of carotenes early in the pathway (León et al., 2005) [9]. Very high levels are also found in fruit colour mutants, such as Pinalate or Cara Cara oranges, resulting from abnormal carotenoid biosynthesis (Rodrigo, 2003)(Alquezar, Rodrigo, & Zacarías, 2008)[10,11]. Moreover, as it can be inferred from the foods analyzed for this study, important amounts of PT and PTF can be present in plant foods also accumulating other carotenoids as a result of “normal” carotenoid biosynthesis”.
[9] León R, Vila M, Hernánz D, Vilchez C. Production of phytoene by herbicide-treated microalgae Dunaliella bardawil in two-phase systems. Biotechnol bioeng 2005, 92:695-701.
[10] Alquezar, B.; Rodrigo, M.J.; Zacarías, L. Regulation of carotenoid biosynthesis during fruit maturation in the red-fleshed orange mutant Cara Cara. Phytochemistry, 2008, 69(10), 1997-2007.
[11] Rodrigo M.J. Characterization of Pinalate, a novel Citrus sinensis mutant with a fruit- specific alteration that results in yellow pigmentation and decreased ABA content. J. Exp. Bot. 2003, 54(383), 727-738.
Some color indicator (°Hue or chlorophyll/carotenoid index-CCI) could be added to the Table 1 to make a comparison/correlation with the PT and PTF content (in a more detailed way than the Table 3).
Reply: We did not carry out any colorometric determination of the foods analyzed because we had a dietary/nutritional point of view for this study. A high intake of fruit and vegetables, but also that these foods should be of a variety of colors is a recommendation being issued by several organizations (i.e. WCRF/AICR and USDA in the Dietary Guidelines for Americans). Thus, the dietary intake of PT and PTF was calculated according to food groups, breaking the fruit and vegetable group down according to colour (white/yellowish, green, red/orange).
- Figure 3: Did the references 33 and 32 (used to make the figure) use the same analysis and quantification parameters? We believe that it is risky and inadequate to compare data from different works in a unique figure.
Reply: Yes, references 33 and 32 [now: 36 and 37](Olmedilla-Alonso conducted those studies that authored two of this manuscript’ authors) use the same Spanish population with the same food consumption data provided by the ENIDE survey and, the same calculations and software were used to assess the dietary intake of carotenoids. Thus, a complete view of the dietary intake of eight carotenoids in a representative sample of the Spanish population can be obtained from these three papers: Beltrán-de-Miguel et al., Int J Food Sci Nutr. 2015; Estévez-Santiago et al., Int J Food Sci Nutr., 2016; the present manuscript). In the previous papers the fruit and vegetables were also grouped according the same three colors.
Line 292 on now reads: “A comparison of the dietary intakes of PT and PTF with the other six main carotenoids we previously assessed in the Spanish population, with the same food consumption data and calculations for the carotenoid dietary intake (Figure 3) provided interesting information: ….”
See also comment 3 above about references to check the possibility to do this comparison and corresponding figure.
Reply: Please, see answer for comment 2 (second part).
- The discussion could be improved to be more assertive and interesting for the reader (not only a comparison with previously local works like the one from Luxembourg). What is the biological importance of the study regarding the carotenoid biosynthetic pathway?. Regarding the nutritional behavior of the Spanish population?. Regarding nutritional and biotechnological aspects related to colorless vs colored carotenoids?
Reply: In an attempt to include the requested comments, at the end of the discussion the following paragraphs have been added: “The presence of PT and PTF in human plasma, milk and in several tissues at levels comparable to the other major carotenoids in humans is already known [7,38,39]. Recently, it has been reported that they account for ~ 25% total carotenoids in adipose tissue [14] and at they are readily detected in the faeces of humans [40,41]. Concerning possible health-promoting biological actions of the colourless carotenoids, there is evidence of different nature indicating that they may be involved in antioxidant and anti-inflammatory, anticarcinogenic actions [41,-48].Thus, these colourless carotenoids could also help promote health as well as the coloured carotenoids typically studied in this regard (lutein, zeaxanthin, β-cryptoxanthin, α-carotene, β-carotene, lycopene). One key difference is that PT and PTF absorb maximally in the UV region of the spectra, what makes them especially interesting in the protection against this radiation by dietary means. This can be harnessed for the biotechnological production of colourless-carotenoid rich products, for instance microalgae, considering that the biosynthesis of coloured carotenoids can be blocked by chemical compounds[9]"
“Being the precursors of virtually all colored carotenoids it might be hypothesized that their levels and percentual intakes could have been considerably higher before the advent of agriculture and then the domestication of crops based on favorable attributes including their color”.
[38] Khachik, F.; Carvalho, L.; Bernstein, P. S.; Muir, G. J.; Zhao, D.-Y.; Katz, N. B. Chemistry, distribution, and metabolism of tomato carotenoids and their impact on human health. Exp. Biol. Med. (Maywood, N.J.), 2002, 227(10), 845–851.
[39] Khachik, F.; Spangler, C. J.; Smith Jr., J. C.; Canfield, L. M.; Steck, A.; Pfander, H. Identification, quantification and relative concentrations of carotenoids and their metabolites in human milk and serum. Anal.Chem., 1997, 69, 1873–1881.
[40] Stinco, C. M.; Benítez-González, A. M.; Meléndez-Martínez, A. J.; Hernanz, D.; Vicario, I. M. Simultaneous determination of dietary isoprenoids (carotenoids, chlorophylls and tocopherols) in human faeces by Rapid Resolution Liquid Chromatography. J. Chromatog. A, 2019, 1583, 63–72.
[41] Rodríguez-Rodríguez, E.; Beltrán-de-Miguel, B.; Samaniego-Aguilar, K. X.; Sánchez-Prieto, M.; Estévez-Santiago, R.; Olmedilla-Alonso, B. Extraction and analysis by HPLC-DAD of carotenoids in human faeces from Spanish adults. Antioxidants, 2020, 9(6), 1–12.
[42] Zhang, C.-R.; Dissanayake, A. A.; Nair, M. G. Functional food property of honey locust (Gleditsia triacanthos) flowers. J. Func. Foods, 2015, 18.
[43] Gijsbers, L.; van Eekelen, H. D. L. M.; de Haan, L. H. J.; Swier, J. M.; Heijink, N. L.; Kloet, S. K.; Rietjens, I. M. C. M. Induction of peroxisome proliferator-activated receptor γ (PPARγ)-mediated gene expression by tomato (Solanum lycopersicum L.) extracts. J. Agric. Food Chem. 2013, 61(14), 3419–3427.
[44] Meléndez-Martínez, A. J.; Nascimento, A. F.; Wang, Y.; Liu, C.; Mao, Y.; Wang, X.-D. Effect of tomato extract supplementation against high-fat diet-induced hepatic lesions. Hepatobiliary Surg. Nutr. 2013, 2(4), 198–208.
[45] Shaish, A.; Harari, A.; Kamari, Y.; Soudant, E.; Harats, D.; Ben-Amotz, A. A carotenoid algal preparation containing phytoene and phytofluene inhibited LDL oxidation in vitro. Plant Foods Human Nutr., 2008, 63(2), 83–86.
[46] Hirsch, K.; Atzmon, A.; Danilenko, M.; Levy, J.; Sharoni, Y. Lycopene and other carotenoids inhibit estrogenic activity of 17-β-estradiol and genistein in cancer cells. Breast Cancer Research and Treatment, 2007, 104(2), 221–230.
[47] Campbell, J. K.; Stroud, C. K.; Nakamura, M. T.; Lila, M. A.; Erdman, J. W. Serum testosterone is reduced following short-term phytofluene, lycopene, or tomato powder consumption in F344 rats. J. Nutr. 2006, 136(11), 2813–2819.
[48] Ben-Dor, A.; Steiner, M.; Gheber, L.; Danilenko, M.; Dubi, N.; Linnewiel, K.; Zick, A.; Sharoni Y.; Levy, J. Carotenoids activate the antioxidant response element transcription system. Mol. Cancer Therapeutics, 2005, 4(1), 177–186.

Reviewer 2 Report
This article presented the major contributors of colorless carotenoids (phytoene and phytofluene) in Spain. The authors also provided daily intakes of them.
Introduction: Although I expected a bit more through the literature review, it provides enough background by incorporating more relevant references.
Materials and Methods:
Fruits and vegetable samples - okay
Extraction and HPLC analysis of carotenoids in foods - okay minus except for line 79 (external calibration - please include what you used).
Subjects - okay
Dietary Carotenoid intake assessment - 24 fruits and 24 vegetables were selected.
Statistics - The authors did not provide any information
Results: line 157 indicates that PT and PTF content were not analyzed, so I would recommend not listing dietary intake as 0.0
line 166 - typo
DIscussion: lines 262 (missing reference)
Author Response
This article presented the major contributors of colorless carotenoids (phytoene and phytofluene) in Spain. The authors also provided daily intakes of them.
Introduction: Although I expected a bit more through the literature review, it provides enough background by incorporating more relevant references.
Reply: Thank you for your comment. We decided to cite review papers (most of the fifteen cited in the introduction) instead or original research papers because all relevant works were referenced in them.
Materials and Methods:
- Extraction and HPLC analysis of carotenoids in foods - okay minus except for line 79 (external calibration - please include what you used).
Reply: The following sentence has been included: “The concentration of each standard was measured spectrophotometrically according to the conditions described in Britton et al [23]. Full standard curves were constructed with five different concentrations for each carotenoid in triplicate. Subsequently, they were injected into the RRLC in triplicate. Both curves showed good linearity (R2= 0.999). The limit of detection ranged between 0.001 µg for PTF and 0.002 µg for PT, while the limit of quantification ranged from 0.002 µg to 0.007 µg for PTF and PT, respectively”.
[23] Britton, G. UV/Visible Spectroscopy. In Carotenoids. Volume 1B: Spectroscopy; Britton, G., Liaaen-Jensen, S., Pfander, H., Eds.; Birkhäuser: Basel, Switzerland, 1995; pp. 13–62.
- Dietary Carotenoid intake assessment - 24 fruits and 24 vegetables were selected.
Reply: Fruits (n=24) and vegetables (n=24) and for some of them, several varieties and colors were considered for the analysis and the carotenoid intake assessment. Now we have modified these data including the number of varieties and of different colors and thus: “Foods considered in the calculations of the dietary intake of phytoene and phytofluene were (n=66): A) fruits (n=33) …… ; B) Vegetables (n=28)….; C) juices and processed foods (n=5)….”
In Results, in lines …., now reads: “Colourless carotenoids were detected in 21 of the 66 foods used in the dietary PT and PTF dietary assessment”.
- Statistics - The authors did not provide any information
Reply: There is no statistical analysis section because no comparisons between groups were made. The mean consumption of foods (g/p/d) obtained from the Spanish National Dietary Survey (ref. 24) was used to calculate the PT and PTF intake (described in lines 146-157).
- Results: line 157 indicates that PT and PTF content were not analyzed, so I would recommend not listing dietary intake as 0.0
Reply: PT or PTF were not analyzed in foods with the superscript b (loquat, orange, pear and tomato sauce). Their contents were compiled from the Database of Carotenoid Contents in Ibero-American Foods published by Dias et al (J Agric Food Chem., 2018). There was data for PT content in loquat, orange and pear, but not for PTF, and now, the 0.0 for the PTF dietary intakes have been removed. There was data for PTF content in tomato sauce, but not for PT, and now, the 0.0, but the PT dietary intake from this tomato sauce has been removed.
- line 166 - typo
Reply: typo has been corrected.
- Discussion: lines 262 (missing reference)
Reply: Reference 15 has now been included in line 262 (Biehler et al., 2012).
